# META-LEARNING WITH LATENT EMBEDDING OPTIMIZATION

**Andrei A. Rusu, Dushyant Rao, Jakub Sygnowski, Oriol Vinyals,**
**Razvan Pascanu, Simon Osindero & Raia Hadsell**
DeepMind, London, UK
{andreirusu, dushyantr, sygi, vinyals,
razp, osindero, raia}@google.com

## ABSTRACT

Gradient-based meta-learning techniques are both widely applicable and proficient at solving challenging few-shot learning and fast adaptation problems. However, they have practical difficulties when operating on high-dimensional parameter spaces in extreme low-data regimes. We show that it is possible to bypass these limitations by learning a data-dependent latent generative representation of model parameters, and performing gradient-based meta-learning in this low-dimensional latent space. The resulting approach, *latent embedding optimization* (LEO), decouples the gradient-based adaptation procedure from the underlying high-dimensional space of model parameters. Our evaluation shows that LEO can achieve state-of-the-art performance on the competitive *mini*ImageNet and *tiered*ImageNet few-shot classification tasks. Further analysis indicates LEO is able to capture uncertainty in the data, and can perform adaptation more effectively by optimizing in latent space.

## 1 INTRODUCTION

Humans have a remarkable ability to quickly grasp new concepts from a very small number of examples or a limited amount of experience, leveraging prior knowledge and context. In contrast, traditional deep learning approaches (LeCun et al., 2015; Schmidhuber, 2015) treat each task independently and hence are often data inefficient – despite providing significant performance improvements across the board, such as for image classification (Simonyan & Zisserman, 2014; He et al., 2016), reinforcement learning (Mnih et al., 2015; Silver et al., 2017), and machine translation (Cho et al., 2014; Sutskever et al., 2014). Just as humans can efficiently learn new tasks, it is desirable for learning algorithms to quickly adapt to and incorporate new and unseen information.

*Few-shot learning tasks* challenge models to learn a new concept or behaviour with very few examples or limited experience (Fei-Fei et al., 2006; Lake et al., 2011). One approach to address this class of problems is *meta-learning*, a broad family of techniques focused on learning how to learn or to quickly adapt to new information. More specifically, *optimization-based* meta-learning approaches (Ravi & Larochelle, 2017; Finn et al., 2017) aim to find a single set of model parameters that can be adapted with a few steps of gradient descent to individual tasks. However, using only a few samples (typically 1 or 5) to compute gradients in a high-dimensional parameter space could make generalization difficult, especially under the constraint of a shared starting point for task-specific adaptation.

In this work we propose a new approach, named *Latent Embedding Optimization (LEO)*, which learns a low-dimensional latent embedding of model parameters and performs optimization-based meta-learning in this space. Intuitively, the approach provides two advantages. First, the initial parameters for a new task are conditioned on the training data, which enables a task-specific starting point for adaptation. By incorporating a relation network into the encoder, this initialization can better consider the joint relationship between all of the input data. Second, by optimizing in the lower-dimensional latent space, the approach can adapt the behaviour of the model more effectively. Further, by allowing this process to be stochastic, the ambiguities present in the few-shot data regime can be expressed.

We demonstrate that LEO achieves state-of-the-art results on both the *mini*ImageNet and *tiered*ImageNet datasets, and run an ablation study and further analysis to show that both conditional parameter generation and optimization in latent space are critical for the success of the method. Source code for our experiments is available at `https://github.com/deepmind/leo`.

## 2 MODEL

### 2.1 PROBLEM DEFINITION

We define the $N$-way $K$-shot problem using the episodic formulation of Vinyals et al. (2016). Each task instance $\mathcal{T}_i$ is a classification problem sampled from a task distribution $p(\mathcal{T})$. The tasks are divided into a *training meta-set* $\mathcal{S}^{tr}$, *validation meta-set* $\mathcal{S}^{val}$, and *test meta-set* $\mathcal{S}^{test}$, each with a disjoint set of target classes (i.e., a class seen during testing is not seen during training). The validation meta-set is used for model selection, and the testing meta-set is used only for final evaluation.

Each task instance $\mathcal{T}_i \sim p(\mathcal{T})$ is composed of a training set $\mathcal{D}^{tr}$ and validation set $\mathcal{D}^{val}$, and only contains $N$ classes randomly selected from the appropriate meta-set (e.g. for a task instance in the training meta-set, the classes are a subset of those available in $\mathcal{S}^{tr}$). In most setups, the training set $\mathcal{D}^{tr} = \left\{ (\mathbf{x}_n^k, y_n^k) \mid k = 1 \ldots K; n = 1 \ldots N \right\}$ contains $K$ samples for each class. The validation set $\mathcal{D}^{val}$ can contain several other samples from the same classes, providing an estimate of generalization performance on the $N$ classes for this problem instance. We note that the validation set of a problem instance $\mathcal{D}^{val}$ (used to optimize a meta-learning objective) should not be confused with the held-out validation meta-set $\mathcal{S}^{val}$ (used for model selection).

### 2.2 MODEL-AGNOSTIC META-LEARNING

Model-agnostic meta-learning (MAML) (Finn et al., 2017) is an approach to optimization-based meta-learning that is related to our work. For some parametric model $f_\theta$, MAML aims to find a single set of parameters $\theta$ which, using a few optimization steps, can be successfully adapted to any novel task sampled from the same distribution. For a particular task instance $\mathcal{T}_i = \left( \mathcal{D}^{tr}, \mathcal{D}^{val} \right)$, the parameters are adapted to task-specific model parameters $\theta'_i$ by applying some differentiable function, typically an update rule of the form:

$$\theta'_i = \mathcal{G}\left( \theta, \mathcal{D}^{tr} \right), \tag{1}$$

where $\mathcal{G}$ is typically implemented as a step of gradient descent on the few-shot training set $\mathcal{D}^{tr}$, $\theta'_i = \theta - \alpha \nabla_\theta \mathcal{L}_{\mathcal{T}_i}^{tr}(f_\theta)$. Generally, multiple sequential adaptation steps can be applied. The learning rate $\alpha$ can also be meta-learned concurrently, in which case we refer to this algorithm as Meta-SGD (Li et al., 2017). During meta-training, the parameters $\theta$ are updated by back-propagating through the adaptation procedure, in order to reduce errors on the validation set $\mathcal{D}^{val}$:

$$\theta \leftarrow \theta - \eta \nabla_\theta \sum_{\mathcal{T}_i \sim p(\mathcal{T})} \mathcal{L}_{\mathcal{T}_i}^{val}\left( f_{\theta'_i} \right) \tag{2}$$

The approach includes the main ingredients of optimization-based meta-learning with neural networks: *initialization* is done by maintaining an explicit set of model parameters $\theta$; the *adaptation procedure*, or "inner loop", takes $\theta$ as input and returns $\theta'_i$ adapted specifically for task instance $\mathcal{T}_i$, by iteratively using gradient descent (Eq. 1); and *termination*, which is handled simply by choosing a fixed number of optimization steps in the "inner loop". MAML updates $\theta$ by differentiating through the "inner loop" in order to minimize errors of instance-specific adapted models $f_{\theta'_i}$ on the corresponding validation set (Eq. 2). We refer to this process as the "outer loop" of meta-learning. In the next section we use the same stages to describe Latent Embedding Optimization (LEO).

### 2.3 LATENT EMBEDDING OPTIMIZATION FOR META-LEARNING

The primary contribution of this paper is to show that it is possible, and indeed beneficial, to decouple optimization-based meta-learning techniques from the high-dimensional space of model parameters. We achieve this by learning a stochastic latent space with an information bottleneck, conditioned on the input data, from which the high-dimensional parameters are generated.

**Algorithm 1** Latent Embedding Optimization

**Require:** Training meta-set $\mathcal{S}^{tr} \in \mathcal{T}$
**Require:** Learning rates $\alpha, \eta$
 1: Randomly initialize $\phi_e, \phi_r, \phi_d$
 2: Let $\phi = \{\phi_e, \phi_r, \phi_d, \alpha\}$
 3: **while** not converged **do**
 4:  **for** number of tasks in batch **do**
 5:    Sample task instance $\mathcal{T}_i \sim \mathcal{S}^{tr}$
 6:    Let $(\mathcal{D}^{tr}, \mathcal{D}^{val}) = \mathcal{T}_i$
 7:    Encode $\mathcal{D}^{tr}$ to $\mathbf{z}$ using $g_{\phi_e}$ and $g_{\phi_r}$
 8:    Decode $\mathbf{z}$ to initial params $\theta_i$ using $g_{\phi_d}$
 9:    Initialize $\mathbf{z}' = \mathbf{z}, \theta_i' = \theta_i$
10:    **for** number of adaptation steps **do**
11:      Compute training loss $\mathcal{L}_{\mathcal{T}_i}^{tr}(f_{\theta_i'})$
12:      Perform gradient step w.r.t. $\mathbf{z}'$:
        $\mathbf{z}' \leftarrow \mathbf{z}' - \alpha\nabla_{\mathbf{z}'}\mathcal{L}_{\mathcal{T}_i}^{tr}(f_{\theta_i'})$
13:      Decode $\mathbf{z}'$ to obtain $\theta_i'$ using $g_{\phi_d}$
14:    **end for**
15:    Compute validation loss $\mathcal{L}_{\mathcal{T}_i}^{val}(f_{\theta_i'})$
16:  **end for**
17:  Perform gradient step w.r.t $\phi$:
    $\phi \leftarrow \phi - \eta\nabla_\phi \sum_{\mathcal{T}_i}\mathcal{L}_{\mathcal{T}_i}^{val}(f_{\theta_i'})$
18: **end while**

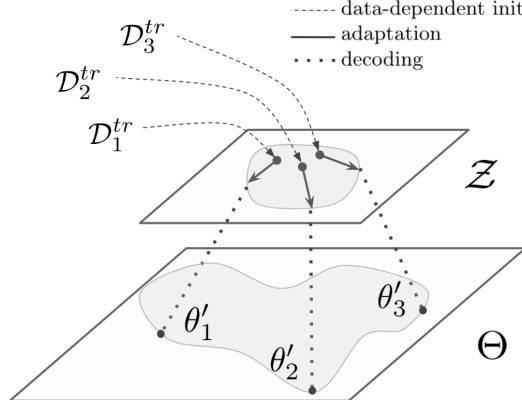

Figure 1: High-level intuition for LEO. While MAML operates directly in a high dimensional parameter space $\Theta$, LEO performs meta-learning within a low-dimensional latent space $\mathcal{Z}$, from which the parameters are generated.

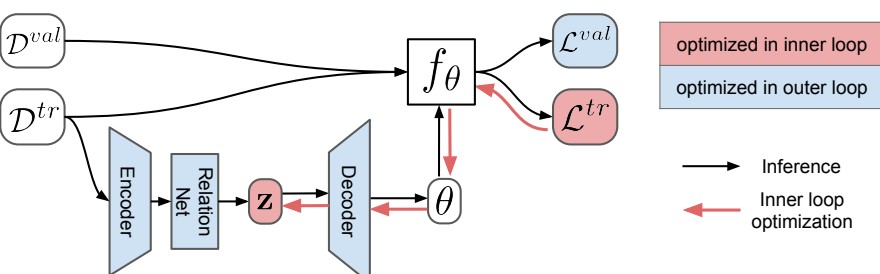

Figure 2: Overview of the architecture of LEO.

Instead of explicitly instantiating and maintaining a unique set of model parameters $\theta$, as in MAML, we learn a generative distribution of model parameters which serves the same purpose. This is a natural extension: we relax the requirement of finding a single optimal $\theta^* \in \Theta$ to that of approximating a data-dependent conditional probability distribution over $\Theta$, which can be more expressive. The choice of architecture, composed of an encoding process, and decoding (or parameter generation) process, enables us to perform the MAML gradient-based adaptation steps (or "inner loop") in the learned, low-dimensional embedding space of the parameter generative model (Figure 1).

### 2.3.1 MODEL OVERVIEW

The high-level operation is then as follows (Algorithm 1). First, given a task instance $\mathcal{T}_i$, the inputs $\{\mathbf{x}_n^k\}$ are passed through a stochastic encoder to produce a latent code $\mathbf{z}$, which is then decoded to parameters $\theta_i$ using a parameter generator[1]. Given these instantiated model parameters, one or more adaptation steps are applied in the latent space, by differentiating the loss with respect to $\mathbf{z}$, taking a gradient step to get $\mathbf{z}'$, decoding new model parameters, and obtaining the new loss. Finally, optimized codes are decoded to produce the final adapted parameters $\theta_i'$, which can be used to perform the task, or compute the task-specific meta-loss. In this way, LEO incorporates aspects of model-based and optimization-based meta-learning, producing parameters that are first conditioned on the input data and then adapted by gradient descent.

---

[1]Note that we omit the task subscript $i$ from latent code $\mathbf{z}$ and input data $\mathbf{x}_n^k$ for clarity.

Figure 2 shows the architecture of the resulting network. Intuitively, the decoder is akin to a generative model, mapping from a low-dimensional latent code to a distribution over model parameters. The encoding process ensures that the initial latent code and parameters before gradient-based adaptation are already data-dependent. This encoding process also exploits a relation network that allows the latent code to be context-dependent, considering the pairwise relationship between all classes in the problem instance. In the following sections, we explain the LEO procedure more formally.

### 2.3.2 INITIALIZATION: GENERATING PARAMETERS CONDITIONED ON A FEW EXAMPLES

The first stage is to instantiate the model parameters that will be adapted to each task instance. Whereas MAML explicitly maintains a single set of model parameters, LEO utilises a data-dependent latent encoding which is then decoded to generate the actual initial parameters. In what follows, we describe an encoding scheme which leverages a relation network to map the few-shot examples into a single latent vector. This design choice allows the approach to consider context when producing a parameter initialization. Intuitively, decision boundaries required for fine-grained distinctions between similar classes might need to be different from those for broader classification.

**Encoding**  The encoding process involves a simple feed-forward mapping of each data point, followed by a relation network that considers the pair-wise relationship between the data in the problem instance. The overall encoding process is defined in Eq. 3, and proceeds as follows. First, each example from a problem instance $\mathcal{T}_i = \left(\mathcal{D}^{tr}, \mathcal{D}^{val}\right) \sim p\left(\mathcal{T}\right)$ is processed by an encoder network $g_{\phi_e} \colon \mathcal{R}^{n_x} \to \mathcal{R}^{n_h}$, which maps from input space to a code in an intermediate hidden-layer code space $\mathcal{H}$. Then, codes in $\mathcal{H}$ corresponding to different training examples are concatenated pair-wise (resulting in $(NK)^2$ pairs in the case of $K$-shot classification) and processed by a relation network $g_{\phi_r}$, in a similar fashion to Oreshkin et al. (2018) and Sung et al. (2017). The $(NK)^2$ outputs are grouped by class and averaged within each group to obtain the $(2 \times N)$ parameters of a probability distribution in a low-dimensional space $\mathcal{Z} = \mathcal{R}^{n_z}$, where $n_z \ll \dim(\theta)$, for each of the $N$ classes.

Thus, given the $K$-shot training samples corresponding to a class $n$: $\mathcal{D}_n^{tr} = \left\{ \left(\mathbf{x}_n^k, y_n^k\right) \mid k = 1 \ldots K \right\}$ the encoder $g_{\phi_e}$ and relation network $g_{\phi_r}$ together parameterize a class-conditional multivariate Gaussian distribution with a diagonal covariance, which we can sample from in order to output a class-dependent latent code $\mathbf{z}_n \in \mathcal{Z}$ as follows:

$$\boldsymbol{\mu}_n^e, \boldsymbol{\sigma}_n^e = \frac{1}{NK^2} \sum_{k_n=1}^{K} \sum_{m=1}^{N} \sum_{k_m=1}^{K} g_{\phi_r} \left( g_{\phi_e} \left(\mathbf{x}_n^{k_n}\right), g_{\phi_e} \left(\mathbf{x}_m^{k_m}\right) \right)$$

$$\mathbf{z}_n \sim q\left(\mathbf{z}_n | \mathcal{D}_n^{tr}\right) = \mathcal{N}\left(\boldsymbol{\mu}_n^e, diag(\boldsymbol{\sigma}_n^{e\,2})\right) \tag{3}$$

Intuitively, the encoder and relation network define a stochastic mapping from one or more class examples to a single code in the latent embedding space $\mathcal{Z}$ corresponding to that class. The final latent code can be obtained as the concatenation of class-dependent codes: $\mathbf{z} = [\mathbf{z}_1, \mathbf{z}_2, \ldots, \mathbf{z}_N]$.

**Decoding**  Without loss of generality, for few-shot classification, we can use the class-specific latent codes to instantiate just the top layer weights of the classifier. This allows the meta-learning in latent space to modulate the important high-level parameters of the classifier, without requiring the generator to produce very high-dimensional parameters. In this case, $f_{\theta_i'}$ is a $N$-way linear softmax classifier, with model parameters $\theta_i' = \left\{ \mathbf{w}_n \mid n = 1 \ldots N \right\}$, and each $\mathbf{x}_n^k$ can be either the raw input or some learned representation[2]. Then, given the latent codes $\mathbf{z}_n \in \mathcal{Z}, n = 1 \ldots N$, the decoder function $g_{\phi_d} \colon \mathcal{Z} \to \Theta$ is used to parameterize a Gaussian distribution with diagonal covariance in model parameter space $\Theta$, from which we can sample class-dependent parameters $\mathbf{w}_n$:

$$\boldsymbol{\mu}_n^d, \boldsymbol{\sigma}_n^d = g_{\phi_d}\left(\mathbf{z}_n\right)$$

$$\mathbf{w}_n \sim p\left(\mathbf{w} | \mathbf{z}_n\right) = \mathcal{N}\left(\boldsymbol{\mu}_n^d, diag(\boldsymbol{\sigma}_n^{d\,2})\right) \tag{4}$$

In other words, codes $\mathbf{z}_n$ are mapped independently to the top-layer parameters $\theta_i$ of a softmax classifier using the decoder $g_{\phi_d}$, which is essentially a stochastic generator of model parameters.

---

[2]As before, we omit the task subscript $i$ from $\mathbf{w}_n$ for clarity.

### 2.3.3 Adaptation by Latent Embedding Optimization (LEO) (The "Inner Loop")

Given the decoded parameters, we can then define the "inner loop" classification loss using the cross-entropy function, as follows:

$$\mathcal{L}_{\mathcal{T}_i}^{tr}\big(f_{\theta_i}\big) = \sum_{(\mathbf{x},y)\in\mathcal{D}^{tr}} \Big[ -\mathbf{w}_y \cdot \mathbf{x} + \log\Big( \sum_{j=1}^{N} e^{\mathbf{w}_j \cdot \mathbf{x}} \Big) \Big] \tag{5}$$

It is important to note that the decoder $g_{\phi_d}$ is a differentiable mapping between the latent space $\mathcal{Z}$ and the higher-dimensional model parameter space $\Theta$. Primarily, this allows gradient-based optimization of the latent codes with respect to the training loss, with $\mathbf{z}'_n = \mathbf{z}_n - \alpha\nabla_{\mathbf{z}_n}\mathcal{L}_{\mathcal{T}_i}^{tr}$. The decoder $g_{\phi_d}$ will convert adapted latent codes $\mathbf{z}'_n$ to effective model parameters $\theta'_i$ for each adaptation step, which can be repeated several times, as in Algorithm 1. In addition, by backpropagating errors through the decoder, the encoder and relation net can learn to provide a data-conditioned latent encoding $\mathbf{z}$ that produces an appropriate initialization point $\theta_i$ for the classifier model.

### 2.3.4 Meta-Training Strategy (The "Outer Loop")

For each task instance $\mathcal{T}_i$, the initialization and adaptation procedure produce a new classifier $f_{\theta'_i}$ tailored to the training set $\mathcal{D}^{tr}$ of the instance, which we can then evaluate on the validation set of that instance $\mathcal{D}^{val}$. During meta-training we use that evaluation to differentiate through the "inner loop" and update the encoder, relation, and decoder network parameters: $\phi_e$, $\phi_r$, and $\phi_d$. Meta-training is performed by minimizing the following objective:

$$\min_{\phi_e,\phi_r,\phi_d} \sum_{\mathcal{T}_i\sim p(\mathcal{T})} \Big[ \mathcal{L}_{\mathcal{T}_i}^{val}\big(f_{\theta'_i}\big) + \beta D_{KL}\big(q(\mathbf{z}_n|\mathcal{D}_n^{tr})||p(\mathbf{z}_n)\big) + \gamma||\text{stopgrad}(\mathbf{z}'_n) - \mathbf{z}_n||_2^2 \Big] + R \tag{6}$$

where $p(\mathbf{z}_n) = \mathcal{N}(0,\mathcal{I})$. Similar to the loss defined in (Higgins et al., 2017) we use a weighted KL-divergence term to regularize the latent space and encourage the generative model to learn a disentangled embedding, which should also simplify the LEO "inner loop" by removing correlations between latent space gradient dimensions. The third term in Eq. (6) encourages the encoder and relation net to output a parameter initialization that is close to the adapted code, thereby reducing the load of the adaptation procedure if possible.

$L_2$ regularization was used with all weights of the model, as well as a soft, layer-wise orthogonality constraint on decoder network weights, which encourages the dimensions of the latent code as well as the decoder network to be maximally expressive. In the case of linear encoder, relation, and decoder networks, and assuming that $\mathcal{C}_d$ is the correlation matrix between rows of $\phi_d$, then the regularization term takes the following form:

$$R = \lambda_1\Big(||\phi_e||_2^2 + ||\phi_r||_2^2 + ||\phi_d||_2^2\Big) + \lambda_2||\mathcal{C}_d - \mathcal{I}||_2 \tag{7}$$

### 2.3.5 Beyond Classification and Linear Output Layers

Thus far we have used few-shot classification as a working example to highlight our proposed method, and in this domain we generate only a single linear output layer. However, our approach can be applied to any model $f_{\theta_i}$ which maps observations to outputs, e.g. a nonlinear MLP or LSTM, by using a single latent code $\mathbf{z}$ to generate the entire parameter vector $\theta_i$ with an appropriate decoder. In the general case, $\mathbf{z}$ is conditioned on $\mathcal{D}^{tr}$ by passing both inputs and labels to the encoder. Furthermore, the loss $\mathcal{L}_{\mathcal{T}_i}$ is not restricted to be a classification loss, and can be replaced by any differentiable loss function which can be computed on $\mathcal{D}^{tr}$ and $\mathcal{D}^{val}$ sets of a task instance $\mathcal{T}_i$.

## 3 Related Work

The problem of few-shot adaptation has been approached in the context of *fast weights* (Hinton & Plaut, 1987; Ba et al., 2016), *learning-to-learn* (Schmidhuber, 1987; Thrun & Pratt, 1998; Hochreiter et al., 2001; Andrychowicz et al., 2016), and through *meta-learning*. Many recent approaches to meta-learning can be broadly categorized as metric-based methods, which focus on learning similarity metrics for members of the same class (e.g. Koch et al., 2015; Vinyals et al., 2016; Snell et al.,

2017); memory-based methods, which exploit memory architectures to store key training examples or directly encode fast adaptation algorithms (e.g. Santoro et al., 2016; Ravi & Larochelle, 2017); and optimization-based methods, which search for parameters that are conducive to fast gradient-based adaptation to new tasks (e.g. Finn et al., 2017; 2018).

Related work has also explored the use of one neural network to produce (some fraction of) the parameters of another (Ha et al., 2016; Krueger et al., 2017), with some approaches focusing on the goal of fast adaptation. Munkhdalai et al. (2017) meta-learn an algorithm to change additive biases across deep networks conditioned on the few-shot training samples. In contrast, Gidaris & Komodakis (2018) use an attention kernel to output class conditional mixing of linear output weights for novel categories, starting from a pre-trained deep model. Qiao et al. (2017) learn to output top linear layer parameters from the activations provided by a pre-trained feature embedding, but they do not make use of gradient-based adaptation. None of the aforementioned approaches to fast adaptation explicitly learn a probability distribution over model parameters, or make use of latent variable generative models to characterize it.

Approaches which use optimization-based meta-learning include MAML (Finn et al., 2017) and REPTILE (Nichol & Schulman, 2018). While MAML backpropagates the meta-loss through the "inner loop", REPTILE simplifies the computation by incorporating an $L_2$ loss which updates the meta-model parameters towards the instance-specific adapted models. These approaches use the full, high-dimensional set of model parameters within the "inner loop", while Lee & Choi (2018) learn a layer-wise subspace in which to use gradient-based adaptation. However, it is not clear how these methods scale to large expressive models such as residual networks (especially given the uncertainty in the few-shot data regime), since MAML is prone to overfitting (Mishra et al., 2018). Recognizing this issue, Zhou et al. (2018) train a deep input representation, or "concept space", and use it as input to an MLP meta-learner, but perform gradient-based adaptation directly in its parameter space, which is still comparatively high-dimensional. As we will show, performing adaptation in latent space to generate a simple linear layer can lead to superior generalization.

Probabilistic meta-learning approaches such as those of Bauer et al. (2017) and Grant et al. (2018) have shown the advantages of learning Gaussian posteriors over model parameters. Concurrently with our work, Kim et al. (2018) and Finn et al. (2018) propose probabilistic extensions to MAML that are trained using a variational approximation, using simple posteriors. However, it is not immediately clear how to extend them to more complex distributions with a more diverse set of tasks. Other concurrent works have introduced deep parameter generators (Lacoste et al., 2018; Wu et al., 2018) that can better capture a wider distribution of model parameters, but do not employ gradient-based adaptation. In contrast, our approach employs both a generative model of parameters, and adaptation in a low-dimensional latent space, aided by a data-dependent initialization.

Finally, recently proposed Neural Processes (Garnelo et al., 2018a;b) bear similarity to our work: they also learn a mapping to and from a latent space that can be used for few-shot function estimation. However, coming from a Gaussian processes perspective, their work does not perform "inner loop" adaptation and is trained by optimizing a variational objective.

## 4 EVALUATION

We evaluate the proposed approach on few-shot regression and classification tasks. This evaluation aims to answer the following key questions: (1) Is LEO capable of modeling a distribution over model parameters when faced with uncertainty? (2) Can LEO learn from multimodal task distributions and is this reflected in ambiguous problem instances, where multiple distinct solutions are possible? (3) Is LEO competitive on large-scale few-shot learning benchmarks?

### 4.1 FEW-SHOT REGRESSION

To answer the first two questions we adopt the simple regression task of Finn et al. (2018). 1D regression problems are generated in equal proportions using either a sine wave with random amplitude and phase, or a line with random slope and intercept. Inputs are sampled randomly, creating a multimodal task distribution. Crucially, random Gaussian noise with standard deviation 0.3 is added to regression targets. Coupled with the small number of training samples (5-shot), the task is challenging for 2 main reasons: (1) learning a distribution over models becomes necessary, in order

to account for the uncertainty introduced by noisy labels; (2) problem instances may be likely under both modes: in some cases a sine wave may fit the data as well as a line. Faced with such ambiguity, learning a generative distribution of model parameters should allow several different likely models to be sampled, in a similar way to how generative models such as VAEs can capture different modes of a multimodal data distribution.

We used a 3-layer MLP as the underlying model architecture of $f_\theta$, and we produced the entire parameter tensor $\theta$ with the LEO generator, conditionally on $D^{tr}$, the few-shot training inputs concatenated with noisy labels. For further details, see Appendix A.

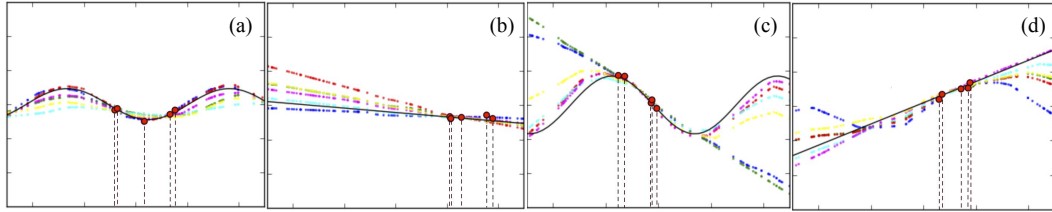

Figure 3: Meta-learning with LEO of a multimodal task distribution with sines and lines, using 5-shot regression with noisy targets. Our model outputs a distribution of possible solutions, which is also multimodal in ambiguous cases. True regression targets are plotted in black, while the 5 training examples are highlighted with red circles and vertical dashed lines. Several samples from our model are plotted with dotted lines (best seen in color).

In Figure 3 we show samples from a single model trained on noisy sines and lines, with true regression targets in black and training samples marked with red circles and vertical dashed lines. Plots (a) and (b) illustrate how LEO captures some of the uncertainty in ambiguous problem instances within each mode, especially in parts of the input space far from any training samples. Conversely, in parts which contain data, models fit the regression target well. Interestingly, when both sines and lines could explain the data, as shown in panels (c) and (d), we see that LEO can sample very different models, from both families, reflecting its ability to represent parametric uncertainty appropriately.

## 4.2 FEW-SHOT CLASSIFICATION

In order to answer the final question we scale up our approach to 1-shot and 5-shot classification problems defined using two commonly used ImageNet subsets.

### 4.2.1 DATASETS

The **_mini_ImageNet** dataset (Vinyals et al., 2016) is a subset of 100 classes selected randomly from the ILSVRC-12 dataset (Russakovsky et al., 2014) with 600 images sampled from each class. Following the split proposed by Ravi & Larochelle (2017), the dataset is divided into training, validation, and test meta-sets, with 64, 16, and 20 classes respectively.

The **_tiered_ImageNet** dataset (Ren et al., 2018) is a larger subset of ILSVRC-12 with 608 classes (779,165 images) grouped into 34 higher-level nodes in the ImageNet human-curated hierarchy (Deng et al., 2009a). This set of nodes is partitioned into 20, 6, and 8 disjoint sets of training, validation, and testing nodes, and the corresponding classes form the respective meta-sets. As argued in Ren et al. (2018), this split near the root of the ImageNet hierarchy results in a more challenging, yet realistic regime with test classes that are less similar to training classes.

### 4.2.2 PRE-TRAINED FEATURES

Two potential difficulties of using LEO to instantiate parameters with a generator network are: (1) modeling distributions over very high-dimensional parameter spaces; and (2) requiring meta-learning (and hence, gradient computation in the inner loop) to be performed with respect to a high-dimensional input space. We address these issues by pre-training a visual representation of the data and then using the generator to instantiate the parameters for the final layer - a linear softmax classifier operating on this representation. We train a 28-layer Wide Residual Network (WRN-28-10) (Zagoruyko & Komodakis, 2016a) with supervised classification using only data and classes

from the training meta-set. Recent state-of-the-art approaches use the penultimate layer representation (Zhou et al., 2018; Qiao et al., 2017; Bauer et al., 2017; Gidaris & Komodakis, 2018); however, we choose the intermediate feature representation in layer 21, given that higher layers tend to specialize to the training distribution (Yosinski et al., 2014). For details regarding the training, evaluation, and network architectures, see Appendix B.

### 4.2.3 FINE-TUNING

Following the LEO adaptation procedure (Algorithm 1) we also use fine-tuning[3] by performing a few steps of gradient-based adaptation directly in parameter space using the few-shot set $\mathcal{D}^{tr}$. This is similar to the adaptation procedure of MAML, or Meta-SGD (Li et al., 2017) when the learning rates are learned, with the important difference that starting points of fine-tuning are custom generated by LEO for every task instance $\mathcal{T}_i$. Empirically, we find that fine-tuning applies a very small change to the parameters with only a slight improvement in performance on supervised classification tasks.

### 4.3 RESULTS

| Model | *mini*ImageNet test accuracy | |
| --- | --- | --- |
| | 1-shot | 5-shot |
| Matching networks (Vinyals et al., 2016) | $43.56 \pm 0.84\%$ | $55.31 \pm 0.73\%$ |
| Meta-learner LSTM (Ravi & Larochelle, 2017) | $43.44 \pm 0.77\%$ | $60.60 \pm 0.71\%$ |
| MAML (Finn et al., 2017) | $48.70 \pm 1.84\%$ | $63.11 \pm 0.92\%$ |
| LLAMA (Grant et al., 2018) | $49.40 \pm 1.83\%$ | - |
| REPTILE (Nichol & Schulman, 2018) | $49.97 \pm 0.32\%$ | $65.99 \pm 0.58\%$ |
| PLATIPUS (Finn et al., 2018) | $50.13 \pm 1.86\%$ | - |
| Meta-SGD (our features) | $54.24 \pm 0.03\%$ | $70.86 \pm 0.04\%$ |
| SNAIL (Mishra et al., 2018) | $55.71 \pm 0.99\%$ | $68.88 \pm 0.92\%$ |
| (Gidaris & Komodakis, 2018) | $56.20 \pm 0.86\%$ | $73.00 \pm 0.64\%$ |
| (Bauer et al., 2017) | $56.30 \pm 0.40\%$ | $73.90 \pm 0.30\%$ |
| (Munkhdalai et al., 2017) | $57.10 \pm 0.70\%$ | $70.04 \pm 0.63\%$ |
| DEML+Meta-SGD (Zhou et al., 2018) [4] | $58.49 \pm 0.91\%$ | $71.28 \pm 0.69\%$ |
| TADAM (Oreshkin et al., 2018) | $58.50 \pm 0.30\%$ | $76.70 \pm 0.30\%$ |
| (Qiao et al., 2017) | $59.60 \pm 0.41\%$ | $73.74 \pm 0.19\%$ |
| **LEO (ours)** | $\mathbf{61.76 \pm 0.08}\%$ | $\mathbf{77.59 \pm 0.12}\%$ |
| Model | *tiered*ImageNet test accuracy | |
| | 1-shot | 5-shot |
| MAML (deeper net, evaluated in Liu et al. (2018)) | $51.67 \pm 1.81\%$ | $70.30 \pm 0.08\%$ |
| Prototypical Nets (Ren et al., 2018) | $53.31 \pm 0.89\%$ | $72.69 \pm 0.74\%$ |
| Relation Net (evaluated in Liu et al. (2018)) | $54.48 \pm 0.93\%$ | $71.32 \pm 0.78\%$ |
| Transductive Prop. Nets (Liu et al., 2018) | $57.41 \pm 0.94\%$ | $71.55 \pm 0.74\%$ |
| Meta-SGD (our features) | $62.95 \pm 0.03\%$ | $79.34 \pm 0.06\%$ |
| **LEO (ours)** | $\mathbf{66.33 \pm 0.05}\%$ | $\mathbf{81.44 \pm 0.09}\%$ |

Table 1: Test accuracies on *mini*ImageNet and *tiered*ImageNet. For each dataset, the first set of results use convolutional networks, while the second use much deeper residual networks, predominantly in conjuction with pre-training.

The classification accuracies for LEO and other baselines are shown in Table 1. LEO sets the new state-of-the-art performance on the 1-shot and 5-shot tasks for both *mini*ImageNet and *tiered*ImageNet datasets. We also evaluated LEO on the "multi-view" feature representation used by Qiao et al. (2017) with *mini*ImageNet, which involves significant data augmentation compared to the approaches in Table 1. LEO is state-of-the-art using these features as well, with $63.97 \pm 0.20\%$ and $79.49 \pm 0.70\%$ test accuracies on the 1-shot and 5-shot tasks respectively.

### 4.4 ABLATION STUDY

To assess the effects of different components, we also performed an ablation study, with detailed results in Table 2. To ensure a fair comparison, all approaches begin with the same pre-trained

---

[3]In this context, "fine-tuning" refers to final adaptation in parameter space, rather than fine-tuning the pre-trained feature extractor.

[4]Uses the ImageNet-200 dataset (Deng et al., 2009b) to train the concept generator.

| Model | *mini*ImageNet test accuracy | | *tiered*ImageNet test accuracy | |
|---|---|---|---|---|
| | 1-shot | 5-shot | 1-shot | 5-shot |
| Meta-SGD (our features) | $54.24 \pm 0.03\%$ | $70.86 \pm 0.04\%$ | $62.95 \pm 0.03\%$ | $79.34 \pm 0.06\%$ |
| Conditional generator only | $60.33 \pm 0.11\%$ | $74.53 \pm 0.11\%$ | $65.17 \pm 0.15\%$ | $78.77 \pm 0.03\%$ |
| Conditional generator + fine-tuning | $60.62 \pm 0.31\%$ | $76.42 \pm 0.09\%$ | $65.74 \pm 0.28\%$ | $80.65 \pm 0.07\%$ |
| Previous SOTA | $59.60 \pm 0.41\%$ | $76.70 \pm 0.30\%$ | $57.41 \pm 0.94\%$ | $72.69 \pm 0.74\%$ |
| LEO (random prior) | $61.01 \pm 0.12\%$ | $77.27 \pm 0.05\%$ | $65.39 \pm 0.10\%$ | $80.83 \pm 0.13\%$ |
| LEO (deterministic) | $61.48 \pm 0.05\%$ | $76.53 \pm 0.24\%$ | $\mathbf{66.18 \pm 0.17\%}$ | $\mathbf{82.06 \pm 0.08\%}$ |
| LEO (no fine-tuning) | $\mathbf{61.62 \pm 0.15\%}$ | $\mathbf{77.46 \pm 0.12\%}$ | $\mathbf{66.14 \pm 0.17\%}$ | $80.89 \pm 0.11\%$ |
| LEO (ours) | $\mathbf{61.76 \pm 0.08\%}$ | $\mathbf{77.59 \pm 0.12\%}$ | $\mathbf{66.33 \pm 0.05\%}$ | $81.44 \pm 0.09\%$ |

Table 2: Ablation study and comparison to Meta-SGD. Unless otherwise specified, LEO stands for using the stochastic generator for latent embedding optimization followed by fine-tuning.

features (Section 4.2.2). The Meta-SGD case performs gradient-based adaption directly in the parameter space in the same way as MAML, but also meta-learns the inner loop learning rate (as we do for LEO). The main approach, labeled as LEO in the table, uses a stochastic parameter generator for several steps of latent embedding optimization, followed by fine-tuning steps in parameter space (see subsection 4.2.3). All versions of LEO are at or above the previous state-of-the-art on all tasks.

The largest difference in performance is between Meta-SGD and the other cases (all of which exploit a latent representation of model parameters), indicating that the low-dimensional bottleneck is critical for this application. The "conditional generator only" case (without adaptation in latent space) yields a poorer result than LEO, and even adding fine-tuning in parameter space does not recover performance; this illustrates the efficacy of the latent adaptation procedure. The importance of the data-dependent encoding is highlighted by the "random prior" case, in which the encoding process is replaced by the prior $p(\mathbf{z}_n)$, and performance decreases. We also find that incorporating stochasticity can be important for *mini*ImageNet, but not for *tiered*ImageNet, which we hypothesize is because the latter is much larger. Finally, the fine-tuning steps only yield a statistically significant improvement on the 5-shot *tiered*ImageNet task. Thus, both the data-conditional encoding and latent space adaptation are critical to the performance of LEO.

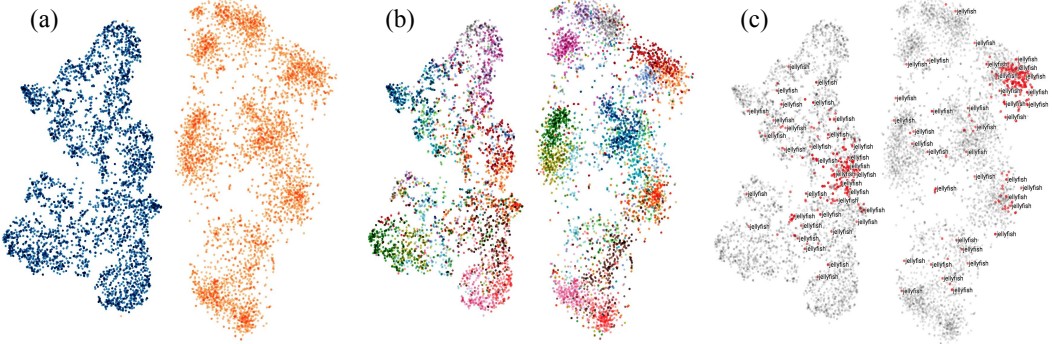

Figure 4: t-SNE plot of latent space codes before and after adaptation: (a) Initial codes $\mathbf{z}_n$ (blue) and adapted codes $\mathbf{z}'_n$ (orange); (b) Same as (a) but colored by class; (c) Same as (a) but highlighting codes $\mathbf{z}_n$ for validation class "Jellyfish" (left) and corresponding adapted codes $\mathbf{z}'_n$ (right).

## 4.5 LATENT EMBEDDING VISUALIZATION

To qualitatively characterize the learnt embedding space, we plot codes produced by the relational encoder before and after the LEO procedure, using a 5-way 1-shot model and 1000 task instances from the validation meta-set of *mini*ImageNet. Figure 4 shows a t-SNE projection of class conditional encoder outputs $\mathbf{z}_n$ as well as their respective final adapted versions $\mathbf{z}'_n$. If the effect of LEO were minimal, we would expect latent codes to have roughly the same structure before and after adaptation. In contrast, Figure 4(a) clearly shows that latent codes change substantially during LEO, since encoder output codes form a large cluster (blue) to which adapted codes (orange) do not belong. Figure 4(b) shows the same t-SNE embedding as (a) colored by class label. Note that encoder

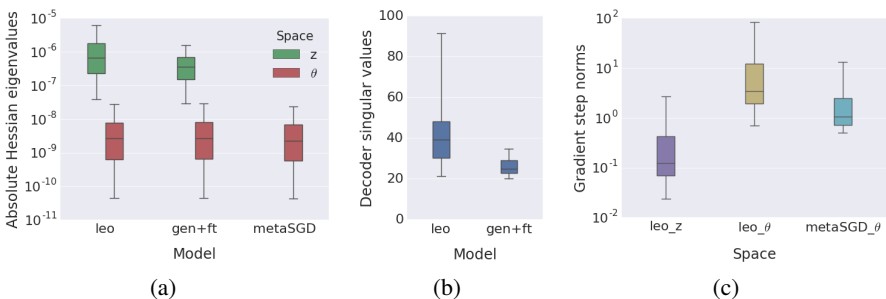

(a)                   (b)                   (c)

Figure 5: Curvature and coverage metrics for a number of different models, computed over 1000 problem instances drawn uniformly from the test meta-set. For all plots, the whiskers span from the $5^{th}$ to $95^{th}$ percentile of the observed quantities.

outputs, on the left side of plot (b), have a lower degree of class conditional separation compared to $\mathbf{z}'_n$ clusters on the right, suggesting that qualitatively different structure is introduced by the LEO procedure. We further illustrate this point by highlighting latent codes for the "Jellyfish" validation class in Figure 4(c), which are substantially different before and after adaptation.

The additional structure of adapted codes $\mathbf{z}'_n$ may explain LEO's superior performance over approaches predicting parameters directly from inputs, since the decoder may not be able to produce sufficiently different weights for different classes given very similar latent codes, especially when the decoder is linear. Conversely, LEO can reduce the uncertainty of the encoder mapping, which is inherent in the few-shot regime, by adapting latent codes with a generic, gradient-based procedure.

### 4.6 Curvature and coverage analysis

We hypothesize that by performing the inner-loop optimization in a lower-dimensional latent space, the adapted solutions do not need to be close together in parameter space, as each latent step can cover a larger region of parameter space and effect a greater change on the underlying function. To support this intuition, we compute a number of curvature and coverage measures, shown in Figure 5.

The curvature provides a measure of the sensitivity of a function with respect to some space. If adapting in latent space allows as much control over the function as in parameter space, one would expect similar curvatures. However, as demonstrated in Figure 5(a), the curvature for LEO in $\mathbf{z}$ space (the absolute eigenvalues of the Hessian of the loss) is 2 orders of magnitude higher than in $\theta$, indicating that a fixed step in $\mathbf{z}$ will change the function more drastically than taking the same step directly in $\theta$. This is also observed in the "gen+ft" case, where the latent embedding is still used, but adaptation is performed directly in $\theta$ space. This suggests that the latent bottleneck is responsible for this effect. Figure 5(b) shows that this is due to the expansion of space caused by the decoder. In this case the decoder is linear, and the singular values describe how much a vector projected through this decoder grows along different directions, with a value of one preserving volume. We observe that the decoder is expanding the space by at least one order of magnitude. Finally, Figure 5(c) demonstrates this effect along the specific gradient directions used in the inner loop adaptation: the small gradient steps in $\mathbf{z}$ taken by LEO induce much larger steps in $\theta$ space, larger than the gradient steps taken by Meta-SGD in $\theta$ space directly. Thus, the results support the intuition that LEO is able to 'transport' models further during adaptation by performing meta-learning in the latent space.

### 5 Conclusions and Future Work

We have introduced Latent Embedding Optimization (LEO), a meta-learning technique which uses a parameter generative model to capture the diverse range of parameters useful for a distribution over tasks, and demonstrated a new state-of-the-art result on the challenging 5-way 1- and 5-shot *mini*ImageNet and *tiered*ImageNet classification problems. LEO achieves this by learning a low-dimensional data-dependent latent embedding, and performing gradient-based adaptation in this space, which means that it allows for a task-specific parameter initialization and can perform adaptation more effectively.

Future work could focus on replacing the pre-trained feature extractor with one learned jointly through meta-learning, or using LEO for tasks in reinforcement learning or with sequential data.

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

## A  EXPERIMENTAL SETUP - REGRESSION

### A.1  REGRESSION TASK DESCRIPTION

We used the experimental setup of Finn et al. (2018) for 1D 5-shot noisy regression tasks. Inputs were sampled uniformly from $[-5, 5]$. A multimodal task distribution was used. Half of the problem instances were sinusoids with amplitude and phase sampled uniformly from $[0.1, 5]$ and $[0, \pi]$ respectively. The other half were lines with slope and intercept sampled uniformly from the interval $[-3, 3]$. Gaussian noise with standard deviation $0.3$ was added to regression targets.

### A.2  LEO NETWORK ARCHITECTURE

As Table 3 shows, the underlying model $f_\theta$ (for which parameters $\theta$ were generated) was a 3-layer MLP with 40 units in all hidden layers and rectifier nonlinearities. A single code $\mathbf{z}$ was used to generate $\theta$ with the decoder, conditioned on concatenated inputs and regression targets from $\mathcal{D}^{tr}$ which were passed as inputs to the encoder. Sampling of latent codes and parameters was used both during training and evaluation.

The encoder was a 3-layer MLP with 32 units per layer and rectifier nonlinearities; the bottleneck embedding space size was: $n_z = 16$. The relation network and decoder were both 3-layer MLPs with 32 units per layer. For simplicity we did not use biases in any layer of the encoder, decoder nor the relation network. Note that the last dimension of the relation network and decoder outputs are two times larger than $n_z$ and $\dim(\theta)$ respectively, as they are used to parameterize both the means and variances of the corresponding Gaussian distributions.

| Part of the model | Architecture | Hidden layer size | Shape of the output |
|---|---|---|---|
| Inference model ($f_\theta$) | 3-layer MLP with ReLU | 40 | $(12, 5, 1)$ |
| Encoder | 3-layer MLP with ReLU | 16 | $(12, 5, 16)$ |
| Relation network | 3-layer MLP with ReLU | 32 | $(12, 2 \times 16)$ |
| Decoder | 3-layer MLP with ReLU | 32 | $(12, 2 \times 1761)$ |

Table 3: Architecture details for 5-way 1-shot *mini*ImageNet and *tiered*ImageNet. The shapes correspond to the meta-training phase. We used a meta-batch of 12 task instances in parallel.

## B  EXPERIMENTAL SETUP - CLASSIFICATION

### B.1  DATA PREPARATION

We used the standard 5-way 1-shot and 5-shot classification setups, where each task instance involves classifying images from 5 different categories sampled randomly from one of the meta-sets, and $\mathcal{D}^{tr}$ contains 1 or 5 training examples respectively. $\mathcal{D}^{val}$ contains 15 samples during meta-training, as decribed in Finn et al. (2017), and all the remaining examples during validation and testing, following Qiao et al. (2017).

We did not employ any data augmentation or feature averaging during meta-learning, or any other data apart from the corresponding training and validation meta-sets. The only exception is the special case of "multi-view" embedding results, where features were averaged over representations of 4 corner and central crops and their horizontal mirrored versions, which we provide for full comparison with Qiao et al. (2017). Apart from the differences described here, the feature training pipeline closely followed that of Qiao et al. (2017).

### B.2  FEATURE PRE-TRAINING

As described in Section 4.2.2, we trained dataset specific feature embeddings before meta-learning, in a similar fashion to Qiao et al. (2017) and Bauer et al. (2017). A Wide Residual Network WRN-28-10 (Zagoruyko & Komodakis, 2016b) with 3 steps of dimensionality reduction was used to clas-

sify images of $80 \times 80$ pixels from only the meta-training set into the corresponding training classes (64 in case of *mini*ImageNet and 351 for *tiered*ImageNet). We used dropout ($p_{keep} = 0.5$) inside residual blocks, as described in (Zagoruyko & Komodakis, 2016b), which is turned off during evaluation and for dataset export. An L2 regularization term of $5e^{-4}$ was used, $0.9$ Nesterov momentum, and SGD with a learning rate schedule. The initial learning rate was $0.1$ and it was multiplied with $0.2$ at the steps given in Table 4. Mini-batches were of size of $1024$. Data augmentation for pre-training was similar to the inception pipeline (Szegedy et al.), with color distortions and image deformations and scaling in training mode. For 64-way evaluation accuracy and dataset export we used only the center crop (with a ratio of $\frac{80}{92}$: about $85.95\%$ of the image) which was then resized to $80 \times 80$ and passed to the network.

| Dataset | Step 1 | Step 2 | Step 3 | Step 4 | Step 5 | Total Steps |
|---|---|---|---|---|---|---|
| *mini*ImageNet | $3 \times 10^3$ | $5 \times 10^3$ | $7 \times 10^3$ | $8 \times 10^3$ | $9 \times 10^3$ | $1 \times 10^4$ |
| *tiered*ImageNet | $2 \times 10^4$ | $2.5 \times 10^4$ | $3 \times 10^4$ | $3.5 \times 10^4$ | $4 \times 10^4$ | $5 \times 10^4$ |

Table 4: Learning rate annealing schedules used to train feature extractors for *mini*ImageNet and *tiered*ImageNet.

Activations in layer 21, with average pooling over spatial dimensions, were precomputed and saved as feature embeddings with $n_x = \dim(\mathbf{x}) = 640$, which substantially simplified the meta-learning process.

### B.3 LEO Network Architecture

We used the same network architecture of parameter generator for all datasets and tasks. The encoder and decoder networks were linear with the bottleneck embedding space of size $n_z = 64$. The relation network was a 3-layer fully connected network with $128$ units per layer and rectifier nonlinearities. For simplicity we did not use biases in any layer of the encoder, decoder nor the relation network. Table 5 summarizes this information. Note that the last dimension of the relation network and decoder outputs are two times larger than $n_z$ and $\dim(\mathbf{x})$ respectively, as they are used to parameterize both the means and variances of the corresponding Gaussian distributions.

The "Meta-SGD (our features)" baseline used the same one-layer softmax classifier as base model.

| Part of the model | Architecture | Shape of the output | When trained? |
|---|---|---|---|
| Feature extractor | WRN-28-10 | $(12, 5, 1, 640)$ | before LEO |
| Encoder | linear | $(12, 5, 1, 64)$ | during outer loop |
| Relation network | 3-layer MLP with ReLU | $(12, 5^v, 2 \times 64)$ | during outer loop |
| Decoder | linear | $(12, 2 \times 640)$ | during outer loop |

Table 5: Architecture details for 5-way 1-shot *mini*ImageNet and *tiered*ImageNet. The shapes correspond to the meta-training phase. We used a meta-batch of 12 task instances in parallel.

### B.4 Optimization

We used a parallel implementation similar to that of Finn et al. (2017), where the "inner loop" is performed in parallel on a batch 12 problem instances for every meta-update. Using a relation network in the encoder has negligible computational cost given that $k^2$ is small in typical $k$-shot learning domains, and the relation network is only used once per problem instance, to get the initial model parameters before adaptation. Within the LEO "inner loop" we perform 5 steps of adaptation in latent space, followed by 5 steps of fine-tuning in parameter space. The learning rates for these spaces were meta-learned in a similar fashion to Meta-SGD (Li et al., 2017), after being initialized to 1 and

---

[v]Function $g_{\phi_r}$ from Eq. (3) is applied 25 times (once for each pair of inputs in $\mathcal{D}^{tr}$) and then averaged into 5 class-specific means and variances.

0.001 for the latent and parameter spaces respectively. We applied dropout independently on the feature embedding in every step, with the probability of not being dropped out $p_{keep}$ chosen (together with other hyperparameters) using random search based on the validation meta-set accuracy.

Parameters of the encoder, relation, and decoder networks as well as per-parameter learning rates in latent and parameter spaces were optimized jointly using Adam (Kingma & Ba, 2014) to minimize the meta-learning objective (Eq. 6) over problem instances from the training meta-set, iterating for up to 100 000 steps, with early stopping using validation accuracy.

Meta-learning objectives can lead to difficult optimization processes in practice, specifically when coupled with stochastic sampling in latent and parameters spaces. For ease of experimentation we clip the meta-gradient, as well as its norm, at an absolute value of 0.1. Please note this was only done for the encoder, relation, decoder networks and learning rates, not the inner loop latent space adaptation gradients.

## B.5 HYPER-PARAMETERS

| Hyperparameter | *mini*ImageNet | | *tiered*ImageNet | |
|---|---|---|---|---|
| | 1-shot | 5-shot | 1-shot | 5-shot |
| $\eta$ (Algorithm 1) | 0.00043653954 | 0.00117573555 | 0.00040645397 | 0.00073469522 |
| $\gamma$ (Eq. (6)) | 1.33365371e−9 | 5.39245830e−6 | 1.24305386e−8 | 3.05077069e−6 |
| $\beta$ (Eq. (6)) | 0.124171967 | 0.0440372182 | 7.10800960e−6 | 0.00188644980 |
| $\lambda_1$ (Eq. (7)) | 0.000108982953 | 3.75922509e−6 | 3.10725285e−8 | 4.90658551e−8 |
| $\lambda_2$ (Eq. (7)) | 303.216647 | 0.00844225971 | 5180.09554 | 0.0081711619 |
| $p_{keep}$ | 0.711524088 | 0.755402644 | 0.644395979 | 0.628325359 |

Table 6: Values of hyperparameters chosen to maximize meta-validation accuracy during random search.

To find the best values of hyperparameters, we performed a random grid search and we choose the set which lead to highest validation meta-set accuracy. The reported performance of our models is an average ($\pm$ a standard deviation) over 5 independent runs (using different random seeds) with the best hyperparameters kept fixed. The result of a single run is an average accuracy over 50000 task instances. After choosing hyperparameters (given in Table 6) we used both meta-training and meta-validation sets for training, in line with recent state-of-the-art approaches, e.g. Qiao et al. (2017).

The evaluation of each of the LEO baselines follow the same procedure; in particular, we perform a separate random search for each of them.

## B.6 TRAINING TIME

Training of LEO took 1-2 hours for *mini*ImageNet and around 5 hours for *tiered*ImageNet on a multi-core CPU (for each of the 5 independent runs). Our approach allows for caching the feature embeddings before training LEO, which leads to a very efficient meta-learning process.

Training of the image extractor was more compute-intensive, taking 5 hours for *mini*ImageNet and around a day for *tiered*ImageNet using 32 GPUs.

## B.7 OVERVIEW OF THE TRAINING PROCEDURE

In summary, there are three stages in our approach to meta-training:

1. In the first stage we use 64-way classification to pre-train the feature embedding only on the meta-training set, hence without the meta-validation classes.
2. In the second stage we train LEO on the meta-training set with early stopping on meta-validation, and we choose the best hyperparameters using random grid search.

3. In the third stage we train LEO again from scratch 5 times using the embedding trained in stage 1 and the chosen set of hyperparameters from stage 2. However, in this stage we meta-learn on embeddings from both meta-train and meta-validation sets, with early-stopping on meta-validation.

While it may not be intuitive to use early stopping on meta-validation in stage 3, it is still a proxy for good generalization since it favors models with high performance on classes excluded during feature embedding pre-training.

## B.8 OVERVIEW OF THE EVALUATION PROCEDURE

The procedure for evaluation is similar to meta-training, except that we disable stochasticity and dropout. Naturally, instead of computing the meta-training loss, the parameters (adapted based on $\mathcal{L}_{\mathcal{T}_i}^{tr}$) are only used for inference on that particular task. That is:

1. A problem instance is drawn from the evaluation meta-set.
2. The few-shot samples are encoded to latent space, then decoded; the means are used to initialize the parameters of the inference model.
3. A few steps of adaptation are performed in latent space, followed (optionally) by a few steps of adaptation in parameter space.
4. The resulting parameters are used as the final adapted model for that particular problem instance.

