# OpenReview forum: "Meta-Learning with Latent Embedding Optimization"
_ICLR.cc/2019/Conference_

### Official Review · AnonReviewer1 · 2018-10-26
**Impressive paper that naturally and effectively extends MAML**

**Rating:** 8
**Confidence:** 5

**Review:**

This work presents an extension of the MAML framework for "learning to learn." This extension changes the space in which "inner-loop" gradient steps are taken to adapt the model to a new task, and also introduces stochasticity. The authors validate their proposed method with regression experiments in a toy setting and few-shot classification experiments on mini- and tiered-Imagenet. The latter are well known and competitive benchmarks in few-shot learning.

The primary innovations that distinguish this work from previous gradient-based approaches to meta-learning (namely MAML) are that (i) the initial set of parameters is data-dependent and drawn from a generative distribution; and (ii) the adaptation of model parameters proceeds in a lower-dimensional latent space rather than in the higher-dimensional parameter space. Specifically, model parameters are generated from a distribution parameterized by an adapted latent code at each adaptation step. I find both of these innovations novel.

The experimental results, in which LEO outperforms the state of the art on two benchmarks derived from ImageNet by "comfortable margins," and the ablation study demonstrate convincingly that these innovations are also significant. I also found the curvature analysis and embedding visualization illuminating of the model's function. My one suggestion would be to test the model on realistic data from beyond the image domain, perhaps on something sequential like language (consider the few-shot PTB setting from Vinyals et al. (2016)). I'm aware anecdotally that MAML struggles with adapting RNNs and I wonder if LEO overcomes that weakness.

The paper is clearly written and I had little difficulty in following the algorithmic details, although I'm sure it helped to be familiar with the convoluted meta-learning and inner-/outer- loop frameworks. I recommend it for publication.

Pros:
- Natural, novel extension to gradient-based meta-learning
- state of the art results on two competitive few-shot benchmarks
- good analysis
- clear writing

Cons:
- realistic, high-dim data is only from the image domain

Minor questions for the authors:
- Relation Networks are computationally intensive, although in few-shot learning the sets encoded are fairly small. Can you discuss the computational cost and training time of the full framework?
- What happens empirically when you generate parameters for more than just the output layer in, eg, your convolutional networks?
- What happens if you try to learn networks from scratch through the meta-learning process rather than pre-training and fine-tuning them? Some of the methods you compare against do so, to my understanding.

---

> ### Author Response · Authors · 2018-11-19
> **Thank you for your comments; we have added some clarifications to the paper**
>
> We thank you for your review and comments, which we look to address below.
>
> - My one suggestion would be to test the model on realistic data from beyond the image domain, perhaps on something sequential like language (consider the few-shot PTB setting from Vinyals et al. (2016)).
>
> Thanks for the suggestion! We are definitely interested in extending this approach to other domains (including RL and sequential tasks), and have clarified this in the future work section.
>
> - Relation Networks are computationally intensive, although in few-shot learning the sets encoded are fairly small. Can you discuss the computational cost and training time of the full framework?
>
> The LEO training process is actually quite short, for example taking 1-2 hours (plus pre-training an embedding) on a multi-core CPU for miniImagenet. We have put this information into the appendix B.6. While the relation net may appear to be computationally intensive, note that it is only performed once per problem instance (for the data-dependent initialization and not for each of the adaptation steps), and can be trivially parallelised on a GPU.
>
> - What happens empirically when you generate parameters for more than just the output layer in, eg, your convolutional networks?
>
> The first step in that direction was taken for the few-shot regression task, where LEO is used to instantiate the parameters of a 3-layer MLP,  yielding good results.
> For few-shot classification, this a promising direction for future work. We anticipate that this will be a more difficult optimization problem but the results on the regression problem are encouraging.
>
> - What happens if you try to learn networks from scratch through the meta-learning process rather than pre-training and fine-tuning them? Some of the methods you compare against do so, to my understanding.
>
> To clarify, the pre-training phase refers to the feature extractor whose output is fed into the softmax classifier, and it is kept fixed. Meta-learning is performed for the parameters of the softmax classifier, and the “fine-tuning” that is specified in Section 4.2.3 refers to additional adaptation directly in parameter space (i.e. the softmax parameters) rather than in the latent space. We have added this information to Section 4.2.3 to make this clear.
>
> Note from the ablation study in Table 2 that we observe only marginal gains from fine-tuning.
>
> Getting rid of the separate pre-training phase and learning everything end-to-end is an interesting problem, however an orthogonal one to the contributions of our work. We haven't tried this yet; our intuition is that, due to different learning dynamics of the feature extractor and the meta-learner, it will be much harder optimization-wise. But obviously, having a broader model space will likely lead to the existence of a better global optimum.

---

### Official Review · AnonReviewer2 · 2018-11-04
**lack of details**

**Rating:** 5
**Confidence:** 3

**Review:**

This paper proposes a latent embedding optimization (LEO) method for meta-learning, in particular, few-shot learning.  The proposed model has three meta components, an encoding network, a relation network, and a decoding network. It claims the contribution is to decouple the optimization-based meta-learning techniques from high-dimensional space of model parameters.

The proposed work focuses on the standard few-shot learning scenario. The notable merit of this work is that it presented the so-far best empirical results. On miniImageNet, it produced 61.76% (1-shot) and 77.59(5-shot) accuracy results. This is quite amazing.

The presentation of the work however lacks sufficient details and motivations, which makes it difficult to judge the proposed model. (1) It is not clear what are the specific architectures and model parameter settings for the encoding, decoding and relation networks.  (2) In Eq.(4), it defines \mu_n^d,\sigma_n^d as the output of the decoding network which takes the single z_n as input. I doubt a single z_n input can provide information on both \mu_n^d,\sigma_n^d. (3) How to use the developed model in the testing phase?

---

> ### Public Comment · (anonymous) · 2018-11-10
> **performance**
>
> I am not the performance-driven guy; but the 61.76% (1-shot) and 77.59(5-shot) accuracy results look really impressive. There is not too many details on how to achieve this yet.

---

> > ### Author Response · Authors · 2018-11-19
> > **Thanks for your interest, could you further clarify your comment?**
> >
> > Thank you for your interest in our paper and for the comment!
> >
> > We hope that the ablation study in Table 2 helps with an incremental implementation of our model, since it measures how each component contributes towards performance; the architecture details, experiments, and hyperparameters to reproduce the approach are provided in the appendices.
> >
> > We are committed to make reproducing our results easy for all readers; could you please glance at the updated manuscript and point out any omissions so that we can clarify them shortly?

---

> ### Author Response · Authors · 2018-11-19
> **Thanks for the feedback; we have added additional details and intuitions**
>
> Thanks for your comments and appreciation of our empirical results! We address the concerns as follows: (1) We have plotted the architecture of LEO in a new diagram; (2) we state that z_n is a 64-dimensional code, similar in semantics to typical VAE latents; (3) we have clarified that the testing phase is identical to training, except for disabling stochastic behavior during few-shot classification, which is not atypical for generative models.
> Due to space constraints, a lengthy description of the underlying implementation is given in the Appendices.
>
> The key changes to the paper are detailed as follows:
>
> (1) While architectures and parameter settings will have a big impact empirically, we felt that they distracted from the reader’s understanding of the algorithm. Thus, as done by recent work, we opted to keep the description of the algorithm clean, to maximize clarity, and show all the details we believe necessary for reproducibility in the Appendix.
>
> Concretely, we give the precise architectures and exact sizes of network layers and outputs in Appendix B.3.:
>
> “We used the same network architecture of parameter generator for all datasets and tasks. The encoder and decoder networks were linear with the bottleneck embedding space of size n_s= 64. The relation network was a 3-layer fully connected network with 128 units per layer and rectifier nonlinearities. For simplicity we did not use biases in any layer of the encoder, decoder nor the relation network. Table 4 summarizes this information.  Note that the last dimension of the outputs of the relation network  and  the  decoder  are  two  times  larger  than n_z and  dim(x) respectively,  as  they are  used  to parameterize both means and variances of the corresponding Gaussian distributions.”
> ----------------
>
> We also added a new intuition diagram of the LEO architecture to the main paper, with some motivation:
>
> “Figure 2 shows the architecture of the resulting network. Intuitively, the decoder is akin to a generative model, mapping from a low-dimensional latent code to a distribution over model parameters. The encoding process ensures that the initial latent code and parameters before gradient-based adaptation are already data-dependent. This encoding process also exploits a relation network that allows the latent code to be context-dependent, considering the pairwise relationship between all classes in the problem instance.”
> --------------
>
>
> (2) Such parameterizations of Gaussian distributions are quite common with generative models, and appear for example in the standard variational autoencoder (see https://arxiv.org/abs/1312.6114 ). For few-shot classification,  z_n is a latent vector with 64 dimensions (as stated in Appendix B.3), and is passed through the decoder to produce the means and variances which parameterize the output distribution over inference model parameters. We highlight the relationship to generative models in the main text for clarity, though it should be pointed out that our model cannot be considered an autoencoder (we do not use a reconstruction loss and the input and output spaces of the encoder/decoder are different: data and parameters respectively).
>
>
> (3) In the meta-testing phase, the procedure is the same as in training, except that instead of computing the (outer) meta-training loss, the adapted parameters are only used to perform a task. This is the same procedure as in MAML and other previous work; we have added this information to Appendix B.8 Overview of the evaluation procedure:
>
> “The procedure for evaluation is similar to meta-training, except that we disable stochasticity and dropout.  Naturally, instead of computing the meta-training loss, the parameters (adapted based on Loss_train) are only used for inference on that particular task. That is:
> 1. A problem instance is drawn from the evaluation meta-set.
> 2. The few-shot samples are encoded to latent space, then decoded;  the means are used to initialize the parameters of the inference model.
> 3. A few steps of adaptation are performed in latent space, followed (optionally) by a few steps of adaptation in parameter space.
> 4.  The resulting parameters are used as the final adapted model for that particular problem instance.”
> --------
>
> We hope that these changes clarify the paper for future readers.

---

> > ### Author Response · Authors · 2018-12-09
> > **Have we successfully addressed the concerns?**
> >
> > Thanks for your constructive comments! We are happy to address any remaining concerns.

---

> > ### Public Comment · (anonymous) · 2018-12-17
> > **ablation study on the effect of the relation net**
> >
> > Hi,
> >
> > May I ask what happens if the relation net is removed? How much will it affect the performance?

---

### Official Review · AnonReviewer3 · 2018-11-04
**More intuitions and insights are required to understand the proposed method**

**Rating:** 6
**Confidence:** 5

**Review:**

This paper presents a new meta-learning framework that learns data-dependent latent space and performs fast adaptation in the latent space. To this end, an encoder that maps data to latent codes and a decoder that maps latent codes to parameters are also learned. Experimental results demonstrate its effectiveness for few-shot learning.

Interestingly, the initialization for adaptation is task-dependent, which is different from conventional MAML-like methods. Furthermore, the results on multimodal few-shot regression seems to show that it works well for multimodal task distribution, which is important for generalization of meta-learning. However, there are quite a few questions that are less clear and require more discussions and insights.

1. Since this work relies heavily on an encoding process and a decoding process, more details and discussions on the design and requirement of the encoder and decoder are necessary. Particularly, the inclusion of a relation network in the encoding process seems ad hoc. More explanations may be helpful.

2. It is less clear why this method can deal with multimodal task distribution as shown in the regression experiment. Is it related to the data-dependent model initialization?

3. It applies meta-learning in a learned latent space, which seems quite related to a recent work, Deep Meta-Learning: Learning to Learn in the Concept Space, Arxiv 2018, where meta-learning is performed in a learned concept space. A discussion on its difference to this prior work seems necessary.

---

> ### Author Response · Authors · 2018-11-19
> **Thanks for the feedback; we have added more intuition, insight, and discussion (1)**
>
> We thank the reviewer for their constructive comments, which we have addressed as follows: (1) we have added clarifications on the architecture of LEO, highlighting the role of the relation network; (2) we added an intuitive explanation for why LEO can express multimodal parameter distributions; (3) we discuss (and compare with) DEML.
>
> Below we detail the changes and further clarifications:
>
> (1) We have added a diagram (Figure 2) in Section 2.3 that characterizes the different steps in the process (including encoding, decoding, adaptation, etc) and have consolidated further intuition and justification into Section 2.3.1: Model Overview. We hope this makes the conceptual design of the LEO network architecture clearer:
>
> “Figure 2 shows the architecture of the resulting network. Intuitively, the decoder is akin to a generative model, mapping from a low-dimensional latent code to a distribution over model parameters. The encoding process ensures that the initial latent code and parameters before gradient-based adaptation are already data-dependent. This encoding process also exploits a relation network that allows the latent code to be context-dependent, considering the pairwise relationship between all classes in the problem instance.”
> --------------
>
> For generality, we have decided to first give an overview of LEO independently of the exact underlying model details, but we make the design of the parametric form of “f” concrete in Section 2.3.2. Decoding. Our parameter generative model needs to only produce the final output layer of a deep model for few-shot classification. This is a typical approach in recent state-of-the-art works, e.g. DEML, Qiao et. al 2017. We give the exact sizes of network layers and outputs in Appendix B.3.
>
> “Without  loss  of  generality,  for  few-shot  classification,  we  can  use  the  class-specific latent codes to instantiate just the top layer weights of the classifier.  This allows the meta-learning in latent space to modulate the important high-level parameters of the classifier, without requiring the generator to produce very high-dimensional parameters.”
> --------------
>
> Relation nets are particularly useful for our problem because they allow us to consider “context” when obtaining a parameter initialization. That is, the latent code and the resulting parameters for a particular class will depend on which other classes are present in the problem instance, making the encoder not only data-, but also context-dependent. Oreshkin et. al (2018) and Sung et al. (2018) have previously exploited this for meta-learning, as we mentioned in section 2.3.1. We have made this argument more explicit in Section 2.3.2 for clarity:
>
> “The  first  stage  is  to  instantiate the model parameters that will be adapted to each task instance. Whereas MAML explicitly maintains a single set of model parameters, LEO utilises a data-dependent latent encoding which is then decoded to generate the actual initial parameters. In what follows, we describe an encoding scheme which leverages a relation network to map the few-shot examples into a single latent vector.  This design choice allows the approach to consider context when producing a parameter initialization. Intuitively, decision boundaries required for fine-grained distinctions between similar classes might need to be different from those for broader classification.”
> --------------
>
> (2) Using a stochastic parameter generative model enables LEO to represent multimodal parameter distributions, in a similar way to how standard generative models can capture multimodal distributions of rich, high-dimensional input data (see VAE: https://arxiv.org/abs/1312.6114 ). This motivates our use of a parameter generator, and we have now highlighted this more clearly in Section 4.1: When meta-learning a clearly bi-modal task distribution, e.g. random sines and lines,
>
> “learning a generative distribution of model parameters should allow several different likely models to be sampled, in a similar way to how generative models such as VAEs can capture different modes of a multimodal data distribution.”
> --------------

---

> > ### Author Response · Authors · 2018-11-19
> > **Thanks for the feedback; we have added more intuition, insight, and discussion (2)**
> >
> > (3) Thank you for drawing our attention to this work. It is relevant and we now cite and compare to it in the related work section and in Table 1. However, it is important to point out that what we call the latent space is fundamentally different than the concept space in Deep Meta-Learning (DEML). In our work, the latent space (while data-dependent) is used to generate the parameters of the model, and can hence be viewed as a compressed representation of the model parameter space. In contrast, DEML uses a deep representation of the data upon which meta-learning is performed. While our work performs adaptation of these latent codes (and hence the resulting higher-dimensional parameter space), the DEML approach adapts parameters of the network directly, whose inputs are learned representation belonging to the concept space of DEML.
> >
> > Actually, the DEML concept space plays a similar role  to our feature embedding space (Section 4.2.2) which we use as an input to the model. Thus, the approaches are orthogonal and could potentially be combined in future work.
> >
> > “Zhou et al. (2018) train a deep input representation, or “concept space”, and use it as input to an MLP meta-learner, but perform gradient-based adaptation directly in its parameter space, which is still comparatively high-dimensional.  As we will show, performing adaptation in latent space to generate a simple linear layer can lead to superior generalization.”
> > ----------------
> >
> > We hope that the aforementioned changes will clarify the details and justification of our approach for future readers.

---

> > ### Comment · AnonReviewer3 · 2018-12-08
> > **Interesting work, deserve further studies. Will source codes be open source?**
> >
> > Thanks for the clarification. Most of my concerns are resolved, and so I increase the rating accordingly.

---

> > > ### Author Response · Authors · 2018-12-09
> > > **We are in the process of open-sourcing our code and embeddings!**
> > >
> > > Thank you for helping us improve the paper! We are in the process of open-sourcing our code and embeddings.

---

### Public Comment · (anonymous) · 2018-10-09
**Question about the meta-training procedure**

Congratulations on your superb result on the miniImageNet benchmark.
From the Appendix A.3, my understanding is that you guys use meta-training and meta-validation sets for meta-training.
To prevent overfitting, early-stopping based on the meta-loss evaluated on meta-validation set is required. I cannot clearly see how such early-stopping can be implemented if meta-validation set used for meta-training.

---

> ### Author Response · Authors · 2018-10-12
> **We will update the text to better reflect our reasoning for this choice**
>
> Thank you for your comment and question! The paper and appendices describe the 3 stages in our meta-learning approach:
> - In the first stage we use 64-way classification to pre-train the feature embedding only on the meta-training set, hence without the meta-validation classes.
> - In the second stage we train LEO on the meta-training set with early stopping on meta-validation, and we choose the best hyperparameters using random grid search.
> - In the third stage we train LEO again from scratch 5 times using the embedding trained in stage 1 and the chosen set of hyperparameters from stage 2. However, in this stage we meta-learn on embeddings from both meta-train and meta-validation sets, with early-stopping on meta-validation.
> While it may not be intuitive to use early stopping on meta-validation in stage 3, it is still a proxy for good generalization, since it favors models with high performance on classes excluded during feature embedding pre-training. We will update the text to better reflect our reasoning for this choice!
> Of course, the meta-test set was not used in any stage for training or selecting our models.

---

### Author Response · Authors · 2018-11-19
**Summary of changes**

We thank the reviewers for their feedback, which has helped to considerably strengthen the paper. We have addressed all of the comments, and the key changes we have made include:

(1) A diagram showing the architecture for LEO, providing a visual representation of how the different components work together.
(2) Additional intuition and discussion in the text to better explain and justify the role of the encoder, decoder, and relation net in our approach.
(3) Strengthening the appendix with more parameter settings and architectural details to further facilitate reproducibility.
(4) Other small clarifications throughout the text to address reviewers’ comments and improve readability.

We hope the above changes will make the paper clearer for future readers.

---

### Meta-Review · Area_Chair1 · 2018-12-18
**Good contribution on meta-learning**

**Confidence:** 5
**Recommendation:** Accept (Poster)

**Metareview:**

This work builds on MAML by (1) switching from a single underlying set of parameters to a distribution in a latent lower-dimensional space, and (2) conditioning the initial parameter of each subproblem on the input data.
All reviewers agree that the solid experimental results are impressive, with careful ablation studies to show how conditional parameter generation and optimization in the lower-dimensional space both contribute to the performance. While there were some initial concerns on clarity and experimental details, we feel the revised version has addressed those in a satisfying way.